# Short- and Mid-Term Results Using a 448 Kilohertz Radiofrequency Stimulation on the Elasticity Plantar Fascia Measured by Quantitative Ultrasound Elastography on Active Healthy Subjects: An Open Controlled Clinical Trial with Three Months of Follow-Up

**DOI:** 10.3390/jcm13237475

**Published:** 2024-12-09

**Authors:** Daniel Aguilar-Núñez, Ana González-Muñoz, Santiago Navarro Ledesma

**Affiliations:** 1Department of Nursing and Podiatry, Faculty of Health Sciences, University of Malaga, Arquitecto Francisco Penalosa 3, Ampliación de Campus de Teatinos, 29071 Malaga, Spain; daguilarn.tic@gmail.com; 2Clinica Actium, Avenida Hernán Núñez de Toledo 6, 29018 Malaga, Spain; agonzalezm@correo.ugr.es; 3Clinical Medicine and Public Health PhD Program, Faculty of Health Sciences, University of Granada, Av. de la Ilustración, 60, 18071 Granada, Spain; 4Department of Physiotherapy, Faculty of Health Sciences, University of Granada, Campus of Melilla, Querol Street, 5, 52004 Melilla, Spain

**Keywords:** plantar fascia, elastography, thermography, viscoelastic, open clinical trial

## Abstract

**Background**: This study is an open clinical trial that included 3 months of follow-up. **Methods**: This study aimed to show the changes that occur in the viscoelastic properties of the PF measured by SEL after the six applications of a 448 kHz capacitive resistive monopolar radiofrequency (CRMR) in active, healthy subjects, immediately before the CRMR intervention (T0), during the two-week CRMR intervention program (T1), after the CRMR intervention program (T2), two weeks after the CRMR intervention program (T3), one month after the CRMR intervention program (T4), and three months after the CRMR intervention program (T5). **Results**: Our results showed that the effects of CRMR on the plantar fascia elasticity may last up to one month in a healthy population after a 3-week treatment program when compared to controls, specifically following the medial process of the calcaneal tuberosity (points 1 and 2). However, there were no changes when analyzing points 3 and 4. These findings are a first step towards understanding the duration of the CRMR effects in the elastic properties of the tissue and therefore how long the benefit may last. **Conclusions**: This study showed that changes in the plantar fascia elasticity measured by SEL have been found after a CRMR intervention protocol, similarly to other structures reported in the literature, such as the patellar tendon or supraspinatus tendon.

## 1. Introduction

The plantar fascia (PF) is a fibrous tissue that plays a crucial role in stabilizing the foot during walking and running. It extends towards the toes in three distinct bands, central, medial, and lateral, and originates from the medial process of the calcaneal tuberosity. The central band is the thickest and most susceptible to pathology and deformities [1,2]. By providing stability to the medial longitudinal arch, the PF is essential to the biomechanics of the foot and lower limb [3].

Plantar fasciitis is one of the most common foot conditions, particularly affecting active adults and athletes. It is estimated to affect 10% to 20% of the population at some point in their lives, accounting for a significant portion of consultations in sports medicine and podiatry [1]. The prevalence is higher among women, individuals with excess weight, and those whose activities involve repetitive overload of the foot, such as runners and workers who stand for long periods [4]. PF can significantly impact patients’ quality of life by limiting mobility and reducing foot functionality [4].

Conservative treatments for PF include rest, physical therapy, non-steroidal anti-inflammatory drugs (NSAIDs), and corticosteroid injections [5]. However, in cases where these approaches do not provide relief, more advanced options, such as radiofrequency, have been explored and have gained popularity in recent years [6]. This treatment harnesses the benefits of heat in relieving pain, reducing inflammation, and promoting tissue healing [6,7]. Frequencies between 300 kHz and 1 MHz are typically used, allowing for deep tissue penetration to target structures such as muscles, tendons, and ligaments [8,9,10]. Capacitive resistive monopolar radiofrequency (CRMR) has seen extensive use in the treatment of musculoskeletal disorders, such as knee osteoarthritis and chronic low back pain [8,10]. This therapeutic approach has been shown to improve the function and quality of life in patients, highlighting its potential for addressing biomechanical alterations in structures like the plantar fascia [6,11,12].

To evaluate the elasticity and other mechanical properties of the PF, strain elastography (SEL) has been introduced as a non-invasive technique [13]. SEL assesses tissue stiffness using Young’s modulus (Y = stress/strain), and while direct stress measurements are not possible, a ratio between a reference area and the area of interest has been proposed to yield reliable results [8,14,15]. Recent studies have demonstrated that PF can be accurately assessed using SEL, even by novice evaluators following a short training period [16].

Our hypothesis is that the application of 448 kHz CRMR can induce lasting changes in the elastic properties of the PF, which can be measured through SEL. These changes would enhance our understanding of the physiological responses that occur in healthy PF tissue over time following CRMR intervention, while also supporting the use of SEL as a reliable tool for assessing tissue quality. Furthermore, by examining the elasticity changes in healthy tissue, essential data to guide future assessment by SEL and therapeutic protocols for pathological conditions such as plantar fasciitis will be established.

The aim of this study is to evaluate the changes in the viscoelastic properties of the PF, as measured by SEL, following six sessions of 448 kHz CRMR (twice weekly) over a three-month follow-up period in healthy active subjects. Measurements will be taken at six time points: immediately before the intervention (T0), during the two-week CRMR intervention program (T1), immediately after the end of the intervention program (T2), two weeks after the intervention (T3), one month after the intervention (T4), and three months after the intervention (T5).

## 2. Materials and Methods

### 2.1. Design

This trial used an open clinical style to conduct a study of 3 months with follow-up. This study was registered in a clinical trial database (clinicaltrials.gov, Accessed on 6 June 2022) with ID NCT05460117.

### 2.2. Setting

The subjects for the trial were registered in a therapeutic clinic in Malaga (Spain). Before participating in the study, verbal and informed written consent was provided by all individuals, with clinical and baseline demographic particulars being collected from them. Formal meetings and informative trial documents were used to convey a comprehensive understanding of the trial to the participants. The study, adhering to the principles of the Declaration of Helsinki, was duly registered on ClinicalTrials.gov under the registration identifier NCT05498987. In line with the standard protocol items of the CONSORT Statement, this study was reported [17,18].

### 2.3. Participants

A total of 23 healthy participants were recruited. To assess all participants, quantitative SEL was used immediately prior to the CRMR intervention (T0), during the two-week CRMR intervention (T1), at the end of the CRMR intervention program (T2), two weeks after the CRMR intervention program (T3), one month after the CRMR intervention program (T4), and three months after the CRMR intervention program (T5). The study reached its goal and ended after the comparisons had been analyzed between pre-intervention and post-intervention groups and after the recruitment of the necessary sample size.

The criteria for inclusion were as follows: (i) not experiencing any lower limb injuries in the 2 years prior to the study, (ii) maintaining a physically active lifestyle, (iii) being within 24 to 42 years of age, and (iv) not being in any discomfort or pain during the evaluation. The criteria to exclude participants were defined in the following manner: any participants presenting (i) neurological disease, (ii) orthopedic disease, or (iii) a painful and inflammatory process that can modify cognitive impairment, hearing, vision, or balance and could affect the ability they have in answering study indications. Verbal and informed written consent was provided by all individuals, with clinical and baseline demographic particulars being collected from them. Formal meetings and informative trial documents were used to ensure the participants have a comprehensive understanding of the trial.

### 2.4. Allocation

For each subject, the non-dominant lower limb is regarded as the control group and the dominant lower limb as the intervention group.

### 2.5. Sample Size Calculation

A strong statistical analysis involving t-tests projected a requisite sample size of n = 19 participants for each group, thereby ensuring a statistical power of 90% with a predefined significance level of α = 0.05. The sample size was determined using the EPIDAT software (version 4.2), a publicly accessible tool designed to support data management for healthcare practitioners and epidemiologists.

### 2.6. Intervention Description

#### 2.6.1. Experimental Group

Patients were given six interventions in the plantar fascia of the dominant lower limb for 3 consecutive weeks using 448 kHz CRMR. The whole plantar fascia was stimulated. The patients were asked to lie face down on an examination table, making sure that their entire foot was protruding beyond the edge of the table to make the measurements easier, and the central band of the plantar fascia was exposed [19]. Capacitive resistive monopolar radiofrequency at 448 kHz was delivered using “INDIBA Activ 8” equipment; Indiba group is the company which manufactures “Indiba active 8”, with a peak power of 200 W and 450 VA. Capacitive (CAP) and resistive (RES) waves were applied using electrodes via a coupling medium. CAP and RES waves were applied using electrodes made of metal. The patient was instructed to communicate the onset of this thermal perception (according to patient feedback on his/her perception of heat, with the patient’s perception expected to be 8 out of 10), which allowed the therapist to apply RES and CAP modes accordingly. Moreover, thermal dosages suggested by the manufacture were followed; for example, the intervention duration for a single session on the heel area was 10 min. The posterior aspect anterior of the quadriceps of the foot treated was strategically positioned as the return electrode. Specifically, the CAP mode was administered for 5 min, followed by the RES mode for 5 min. In a continuous wave for 5 min, the RES mode was delivered as an applied thermal dose (hyperthermia) [20].

#### 2.6.2. Control Group

The control group (non-dominant lower limb) did not receive any intervention with 448 kHz CRMR in the plantar fascia.

#### 2.6.3. SEL Measurements

The SECA 804, which is a precise portable electronic scale, was employed to measure the weight of the participants, and the SECA mBCA 555, which is a precise electronic scale, was used to measure their height [12,20]. All US imaging was carried out with an ultrasound GE Logiq-S7 and a 3.0–10.0 MHz linear-array transducer (GE Healthcare, Milwaukee, WI, USA) with an 8 MHz frequency, and “Coded Harmonic Imaging” was employed for the study period. A professional in using ultrasound images of the musculoskeletal conducted probe, who had 10 years of experience, carried out all measurements. All CRMR interventions were administered by a separate, trained specialist to maintain consistency and minimize evaluator bias. The US operator could change the gain, depth, or focus as necessary. The diagnosis or lack of diagnosis of the pathology of plantar fascia was conducted via ultrasound, measuring the thickness of the tissue and establishing the diagnosis based on this quantitative value. According to previous studies, a PF thickness over 4 mm was considered a validated parameter for an acute state of plantar fasciitis and the reference standard for sonographic diagnosis [21]. Participants were positioned with their feet hanging beyond the table edge and in ventral decubitus on a flat table [20]. The participants’ ankles were allowed to relax in a semi-flexed position, and the central band of the foot’s plantar fascia was exposed. According to the ultrasound thickness measurement protocol [21], the plantar fascia thickness was measured strictly vertically in a longitudinal plane over the calcaneus at the thickest part of the fascia, which is primarily at the medial tubercle of the calcaneus [20]. Ultrasound elastography measurements were taken from the calcaneal insertion of the plantar fascia. Four different measurement points were used to evaluate the plantar fascia. Intra-observer variation was minimized by calculating the mean at each of these points.

The stiffness color scheme was red (hard), green (medium), and blue (soft). The color histogram was analyzed, and subsequently, the mean intensity of each color component of the pixels within a standardized area was computed. As indicated by the manufacturer guidelines, the exact raw strain value was calculated from a 5 mm circle in a supple region of the selected area [11,21,22]. Following the manufacturer’s recommendations, the approved compression size was assessed by means of a quality control incorporated into the software. It consists of green bars displayed on a screen, with five bars being the most suitable and one the least suitable. The measured images were of the highest quality, as only sequences with green bars were included.

The SEL values obtained ranged from 0 to 6, where 0 means the softest and 6 the hardest tissue. In addition, there is a representative bar scale with the hardest tissue being shown in red and the softest tissue shown in blue, as well as by the letter “S” (soft) on the bar scale’s top and by the letter “H” (hard) on the bar scale’s bottom. (Figure 1).

### 2.7. Data Analysis

All the analyses used SPSS^®^ Statistics version 21.0 (IBM, Chicago, IL, USA). Data distribution normality was verified using the Shapiro–Wilk test. To analyze both groups, in regard to clinical characteristics, a six-way repeated measurement ANOVA was conducted immediately before the CRMR intervention (T0), during the two-week CRMR intervention (T1), immediately after the CRMR intervention (T2), two weeks after the CRMR intervention (T3), one month after the CRMR intervention (T4), and three months after the CRMR intervention (T5), with three levels corresponding to each time of assessment (T0, T1, T2, T3, T4, and T5). A *p*-value < 0.05 was considered statistically significant. The following calculations were included: Bonferroni adjustments for multiple comparisons were studied. The differences between both group effect sizes, for all quantitative variables, were measured by employing the Cohen d coefficient. The effect sizes that were greater than 0.8 were considered large, those around 0.5 were considered moderate, and those less than 0.2 were considered small.

## 3. Results

During treatment, the experimental group showed dermatological effects, for example, flushing in response to the hyperthermia that was used. In contrast, the control group showed no response and, due to the sham intervention, with no change in temperature, was thought not to be injured. No participant was injured during the study period. To inform participants, sheets with trial information and formal meetings were employed (Figure 2).

### 3.1. Sample Characteristics

In Table 1 demographic characteristics were shown. There were no significant differences in height, weight, age, gender, and fascia plantar elasticity between the compared lower limbs.

### 3.2. Differences in Plantar Fascia Elasticity with a Three-Month Follow-Up

In Table 2, comparisons between groups are described in detail. There were statistically significant differences in the plantar fascia elasticity observed when analyzing changes from one month after the CRMR intervention program (T4) in fascia elasticity points 1 (0.53 Young’s modulus *p* = 0.019) and 2 (0.65 Young’s modulus *p* = 0.01).

## 4. Discussion

This study aimed to show the changes that occur in the viscoelastic properties of the PF measured by SEL after the six applications of a 448 kHz CRMR in active healthy participants and how these changes last over time, specifically tested through one- and three-month follow-ups. Patients received six interventions in the plantar fascia of the dominant lower limb for 3 weeks using 448 kHz CRMR. The results from the present study show that the effects of CRMR on the plantar fascia elasticity may last up to one month in a healthy population after a 3-week treatment program when compared to the controls, specifically following the medial process of calcaneal tuberosity (See Figure 1, points 1 and 2). However, there were no elastic changes when analyzing points 3 and 4.

To the best of our knowledge, this study is the first to investigate the effects of 448 kHz capacitive resistive monopolar radiofrequency (CRMR) on the viscoelastic properties of the plantar fascia (PF) using strain elastography (SEL). This makes direct comparisons with other studies challenging. However, the existing literature on CRMR applied to tendinous and musculoskeletal structures provides some context for interpreting our findings. Previous studies have explored the effects of CRMR on tendons, particularly focusing on structures such as the patellar and supraspinatus tendons [6,11,20]. For example, Navarro-Ledesma et al. [14] demonstrated significant changes in the elasticity of the supraspinatus tendon following CRMR treatment in professional badminton players, with improvements in tendon stiffness lasting up to three months. This aligns with our finding that CRMR can induce lasting changes in tissue elasticity for up to one month, although the supraspinatus tendon is part of the upper extremity, which undergoes different mechanical loading than the PF. Similarly, studies on the patellar tendon have shown immediate thermal changes post-CRMR application, though changes in elasticity were inconsistent when compared to the controls [6,7,11,15]. These findings suggest that while CRMR can generate physiological effects in tendons, the magnitude and duration of these effects may vary based on factors such as tissue type, mechanical demands, and anatomical location.

In our study, we observed significant changes in PF elasticity specifically at points 1 and 2, which correspond to the medial calcaneal insertion and mid-proximal body of the PF. These regions are subject to substantial mechanical loading during gait, which may explain their responsiveness to CRMR treatment [4,23]. The fact that points 3 and 4 did not exhibit significant changes could be attributed to lower mechanical stress in these areas or differences in tissue composition [4]. This selective response in high-demand regions of the PF underscores CRMR’s potential for targeting tissue elasticity enhancements where the mechanical load is highest, which could be crucial for both preventive and rehabilitative applications [7,11], where CRMR produced localized thermal and elastic effects depending on the anatomical site treated. This approach allows clinicians and researchers to obtain precise, reproducible data that may facilitate treatment monitoring and personalized rehabilitation plans. A key distinction between our study and prior research is the use of SEL to quantify tissue elasticity. Most studies on CRMR have relied on subjective or indirect measures of treatment efficacy, such as patient-reported pain scores or functional assessments [12,15,24]. By incorporating SEL, we were able to objectively measure changes in tissue stiffness, providing more precise insights into the biomechanical adaptations induced by CRMR. Our results highlight the potential of SEL to capture subtle changes in tendon and fascia elasticity that may not be detectable through traditional clinical assessments. The differences between our results and those of studies involving other tendons, such as the patellar and supraspinatus tendons [10,11,14,22], may also be due to the distinct biomechanical roles and loading patterns of these structures. The PF, being a load-bearing structure of the lower limb, is subject to continuous mechanical stress during walking and running [25,26], unlike the supraspinatus tendon, which is primarily involved in upper limb movement. This may account for the localized improvements in elasticity at the PF’s insertional points, which are areas of high mechanical demand. Our findings suggest that CRMR could be particularly beneficial in enhancing resilience in high-stress regions of the PF, thus supporting its potential as a therapeutic modality in sports medicine and for populations exposed to repetitive loading. Overall, while our findings are consistent with those of previous studies regarding the general efficacy of CRMR in enhancing tissue elasticity, the specific effects on the PF are novel [27]. Future studies should aim to validate these findings in clinical populations with PF-related pathologies, such as plantar fasciitis, to explore whether the elasticity improvements observed here extend to degenerative tissues. Additionally, studies could investigate the optimal frequency and intensity of CRMR treatments to sustain these benefits long term, further exploring SEL as a measurement standard in pathological settings. Our study provides important insights into the physiological responses of the PF to CRMR. The significant changes in elasticity at points 1 and 2, located at the calcaneal insertion and the mid-proximal body of the PF, are particularly noteworthy, as these areas are highly susceptible to mechanical loading and injury, especially in conditions like plantar fasciitis [7,25,26]. The localized nature of these changes suggests that CRMR may selectively influence areas of the fascia that bear the greatest mechanical stress, possibly enhancing their ability to withstand repetitive loading. The elasticity improvements observed at these points may be related to several mechanisms. CRMR is known to induce hyperthermia in deep tissues, which promotes vasodilation, increases blood flow, and enhances the delivery of oxygen and nutrients to the affected area [23,26,27]. These effects, in turn, may stimulate cellular processes involved in tissue repair and regeneration, such as collagen synthesis and the modulation of inflammatory responses [7]. Given that the PF is composed of dense connective tissue [3,20], which typically exhibits limited regenerative capacity, the ability of CRMR to induce such changes is clinically significant, as it may help counteract the degenerative processes often seen in chronic conditions like plantar fasciitis.

Interestingly, no significant changes were detected at points 3 and 4, which represent the mid-distal and distal portions of the PF. These regions are typically subject to less mechanical stress compared to the calcaneal insertion, which may explain their lack of responsiveness to CRMR [3,26]. Another possible explanation is that the tissue composition and biomechanical properties of the PF vary along its length, with the proximal portions being more fibrous and prone to degeneration, while the distal portions may exhibit greater elasticity and thus be less susceptible to CRMR-induced changes [27]. The absence of changes at these points raises important questions about the specificity of CRMR’s effects on tendinous structures [7,11]. It is possible that CRMR’s ability to induce elastic changes is contingent on the mechanical loading patterns of the tissue being treated [7,11]. This hypothesis is supported by studies on the Achilles and patellar tendons, which have shown that CRMR’s effects are more pronounced in regions of high mechanical demand. Further research is needed to determine whether this pattern holds true for other tendons and whether the absence of changes in the distal PF has clinical implications. The improvements in PF elasticity observed in this study may also have functional consequences. Increased tissue elasticity is generally associated with improved shock absorption and load distribution, which could help protect the PF from further injury [26]. This is particularly relevant in active populations, where repetitive loading of the PF during activities like running and jumping can lead to overuse injuries. By enhancing the viscoelastic properties of the PF, CRMR may help improve the tissue’s resilience to mechanical stress, potentially reducing the risk of conditions such as plantar fasciitis.

### 4.1. Clinical Significance of the Results

The clinical significance of these findings lies in the potential for CRMR to serve as a therapeutic tool for preventing and managing PF-related pathologies. The improvements in tissue elasticity observed at key points of the PF suggest that CRMR could be used to enhance the load-bearing capacity of the fascia, making it more resilient to mechanical stress. This is particularly relevant in sports medicine, where athletes are prone to developing overuse injuries such as plantar fasciitis due to repetitive loading of the PF during training and competition. Additionally, the fact that the effects of CRMR persisted for up to one month post-treatment indicates that this modality may provide longer lasting benefits compared to traditional therapies, which often require frequent sessions to maintain their effects. This could reduce the overall treatment burden for patients and allow for more sustained improvements in tissue function. Another important clinical application of these findings is the potential use of CRMR in injury prevention. By improving PF elasticity in healthy individuals, CRMR may help prevent the onset of plantar fasciitis or other degenerative conditions in populations at risk, such as runners, dancers, and individuals who stand for long periods. The ability to modulate tissue elasticity through non-invasive means opens up new avenues for both prophylactic treatment and rehabilitation.

### 4.2. Strengths and Weaknesses of the Study

One of the primary strengths of this study is its use of SEL, an objective and non-invasive technique that allows for the precise quantification of tissue elasticity. This provides a more accurate assessment of the effects of CRMR on the PF compared to subjective measures such as pain scores or functional tests. Additionally, the study design, which included a three-month follow-up period, allowed us to assess both the short- and medium-term effects of CRMR, providing valuable insights into the duration of its therapeutic benefits. Another strength is the homogeneity of the study population. By focusing on healthy, active individuals with no pre-existing foot conditions, we were able to isolate the effects of CRMR on the PF without the confounding influence of pathology. This is particularly important when studying a structure like the PF, which is prone to degeneration in pathological conditions like plantar fasciitis. Furthermore, another strength of the study is that the ultrasound evaluations were conducted by a single, highly experienced professional, ensuring diagnostic consistency across participants. However, there are several limitations to this study that must be acknowledged. First, the CRMR interventions were administered by a separate specialist, which, while reducing evaluator bias, may introduce a slight limitation regarding consistency in intervention application. Future studies could consider using the same professional for both procedures to further streamline the protocol. Second, the study was conducted exclusively on healthy individuals, limiting the generalizability of the findings to clinical populations with PF-related conditions. Third, the relatively small sample size may limit the statistical power of the findings, particularly when assessing changes at the distal points of the PF, where no significant differences were detected. Larger studies are needed to confirm our results and to explore whether CRMR has differential effects on different regions of the PF. Fourth, the lack of a long-term follow-up beyond three months limits our understanding of the persistence of CRMR’s effects. Finally, the use of the non-dominant limb as the control and the dominant limb as the intervention within the same participants should discussed. This design was selected to control for external individual factors that might influence outcomes, such as nutrition, mental health, and lifestyle habits. By using the same subjects as their own control, we aimed to reduce external variability and ensure that changes in the plantar fascia’s elasticity could be attributed to the CRMR treatment. However, we acknowledge that this design limits the comparison between independent groups, which might provide a more objective perspective by alternating the dominant and non-dominant limbs as the control and intervention across two groups of participants. This limitation should be considered when interpreting the results, and future studies could benefit from an independent group design to further validate and expand our findings.

## 5. Conclusions

The application of a 448 kHz CRMR stimulation program produced measurable changes in the elastic properties of the PF as assessed by SEL, with effects lasting up to one-month post-intervention. These findings represent a foundational step in understanding the potential of CRMR to influence tissue elasticity in load-bearing structures like PF, which has implications for injury prevention, performance enhancement, and rehabilitation strategies in both athletic and general populations. The observed changes in PF elasticity, particularly in high-stress areas, suggest that CRMR may offer a targeted, non-invasive approach for enhancing tissue resilience in populations at risk for overuse injuries, such as athletes and individuals who engage in repetitive loading activities. However, further research is required to confirm these results in clinical populations with PF-related conditions and to determine the optimal frequency, duration, and intensity of CRMR for sustained therapeutic effects. More studies involving patients with plantar pathologies, alongside longitudinal and controlled designs, are necessary to corroborate our findings. Additionally, standardized intervention protocols are essential for deepening our understanding of CRMR’s impact on PF elasticity and for establishing its role in clinical practice.

## Figures and Tables

**Figure 1 jcm-13-07475-f001:**
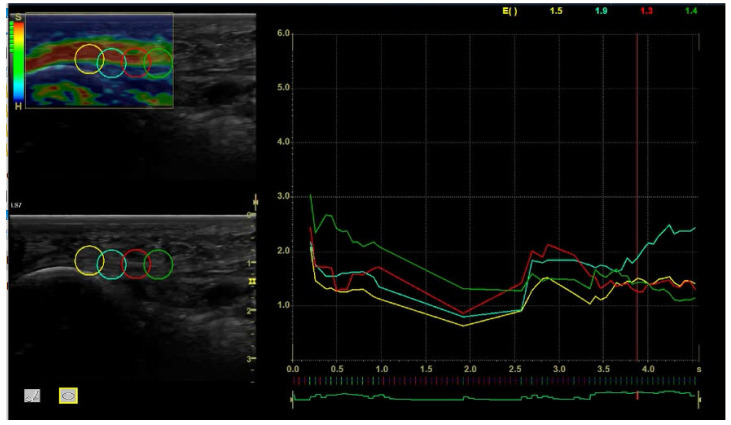
Region of interest (ROI) and SEL measurements. Note: left to right: point 1: PF insertion to the calcaneus; point 2: body of the plantar fascia in the mid-proximal portion; point 3: body of the plantar fascia in the middle portion; point 4: body of the plantar fascia in the mid-distal portion.

**Figure 2 jcm-13-07475-f002:**
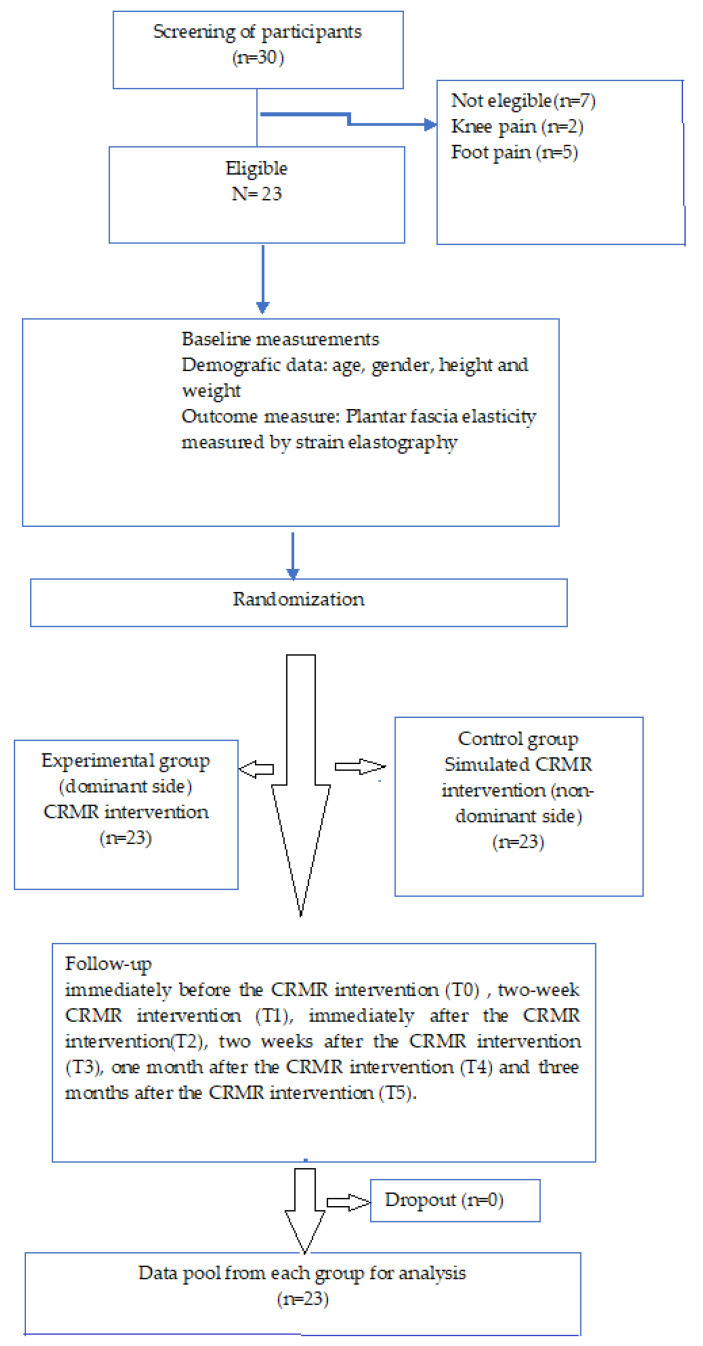
Flow diagram of participants. Note. Description of the study.

**Table 1 jcm-13-07475-t001:** Baseline demographic characteristics.

	Intervention Group (CRMR) (n = 23)	Control Group (Placebo CRMR) (n = 23)
Age (years), mean (SD)	32.7 (10.6)	32.7 (10.6)
Height (cm), mean (SD)	175 (8.51)	175 (8.51)
Weight (kg), mean (SD)	72 (10.4)	72 (10.4)
Plantar fascia elasticity point 1, mean Young’s modulus (SD)	1.56 (0.63)	2.48 (1.55)
Plantar fascia elasticity point 2, mean Young’s modulus (SD)	3.28 (5.33)	2.57 (1.5)
Plantar fascia elasticity point 3, mean Young’s modulus (SD)	2.18 (0.95)	2.27 (0.9)
Plantar fascia elasticity point 4, mean Young’s modulus (SD)	2.67 (1.15)	2.9 (0.75)

Note. Shows demographic characteristics. CRMR: Capacitive resistive monopolar radiofrequency. SD: Standard deviation.

**Table 2 jcm-13-07475-t002:** Between-group differences on plantar fascia elasticity at baseline (T0), during the two-week CRMR intervention (T1), immediately after the CRMR intervention (T2), two weeks after the CRMR intervention (T3), one month after the CRMR intervention (T4), and three months after the CRMR intervention (T5) (95%CI).

	T0 (Immediately Before the CRMR Intervention)	T1 (Two-Week CRMR Intervention)	T2 (Immediately After the CRMR Intervention)	T3 (Two Weeks After CRMR Intervention)	T4 (One Month After CRMR Intervention)	T5 (Three Months After CRMR Intervention)
Plantar fascia elasticity point 1 (mean Young’s modulus)	0.23 *p* = 0.85	−0.5 *p* = −0.21	−0.27 *p* = 0.66	0.11 *p* = 0.99	0.53 *p* = 0.019	−0.52 *p* = 0.29
Plantar fascia elasticity point 2 (mean Young’s modulus)	0.18 *p* = 0.95	0.1 *p* = 1	−0.18 *p* = −0.89	0.30 *p* = 0.61	0.65 *p* = 0.01	0.35 *p* = 0.64
Plantar fascia elasticity point 3 (mean Young’s modulus)	0.2 *p* = 0.91	0.34 *p* = 1	−0.2 *p* = −0.85	0.34 *p* = 0.66	−0.14 *p* = −0.99	−0.16 *p* = 0.97
Plantar fascia elasticity point 4 (mean Young’s modulus)	0.47 *p* = 0.54	−0.01 *p* = −1	−0.49 *p* = 0.25	0.43 *p* = 0.47	0.01 *p* = 1	−0.05 *p* = 1

Note. CRMR= Capacitive resistive monopolar radiofrequency. Statistically significant differences (*p* = 0.05). SE: Size effect. a 95% CI.

## Data Availability

The data are available to consult.

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
