# Peer review of "Short- and Mid-Term Results Using a 448 Kilohertz Radiofrequency Stimulation on the Elasticity Plantar Fascia Measured by Quantitative Ultrasound Elastography on Active Healthy Subjects: An Open Controlled Clinical Trial with Three Months of Follow-Up"

_jcm, 2024, doi:10.3390/jcm13237475_

Round 1
Reviewer 1 Report
Comments and Suggestions for Authors The paper addresses a relevant topic, providing a therapeutic approach with a trending procedure for a very common pathology, which gives it significant scientific and clinical interest. However, I believe the methodology is neither clear nor repeatable. Therefore, I recommend the following to the authors: · Provide a detailed explanation of the inclusion and exclusion criteria. · Clarify how the diagnosis of the pathology was performed and how the "healthy subject" status was confirmed. · Describe the therapeutic protocols with precision. In the current reading, there seems to be heterogeneity in the therapeutic approach, which could bias the results. · When stating that subjects gave both written and verbal consent, clarify whether some provided written consent and others verbal, or if both types of consent were required from all subjects. If both were required, it is unnecessary to specify verbal consent. · Detail the equipment and tools used. · Specify whether there were multiple evaluators or if tasks were performed independently by each evaluator. This could be seen as a positive factor if intra-evaluator practice does not interfere; otherwise, it may highlight the need for experience in the area. Finally, it is unclear whether a total of 23 or 46 subjects were evaluated, as there is a discrepancy between the text and the flowchart.
Author Response
Dear Editor,
We would like to express our sincere gratitude for the opportunity to revise and improve our manuscript titled " Short-and mid-Term results using of 448 kilohertz radiofrequency stimulation on the elasticity plantar fascia measured by quantitative ultrasound elastography on active healthy subjects: an open controlled clinical trial with three months of follow-up". The feedback from the reviewers has been invaluable in enhancing the clarity, rigor, and clinical relevance of our work. We have carefully addressed each of their suggestions and made substantial improvements to the manuscript based on their insightful comments.
We believe that these revisions have significantly strengthened the quality of the paper, and we hope it now meets the standards for publication in your esteemed journal. Thank you for considering our revised submission.
To REVIEWER 1
Comment 1:
The paper addresses a relevant topic, providing a therapeutic approach with a trending procedure for a very common pathology, which gives it significant scientific and clinical interest. However, I believe the methodology is neither clear nor repeatable. Therefore, I recommend the following to the authors: · Provide a detailed explanation of the inclusion and exclusion criteria. ·
Response 1:
The authors have provided a clear and detailed explanation of the inclusion and exclusion criteria following the reviewer´s suggestions.
Comment 2:
Clarify how the diagnosis of the pathology was performed and how the "healthy subject" status was confirmed. ·
Response 2:
The diagnosis of the pathology was confirmed by ultrasound diagnosis. Subjects with a plantar fascia thickness greater than 0.4 cm were amended as potentially pathological according to current literature.
Comment 3:
Describe the therapeutic protocols with precision.
Response 3: Therapeutics protocols have been described with more clarity. We have added more information about the intervention description with 448kHz CRMR and SEL measurements.
Comment 4:
In the current reading, there seems to be heterogeneity in the therapeutic approach, which could bias the results. ·
Response 4: The authors have added more details about the therapeutic approach.
Comment 5:
When stating that subjects gave both written and verbal consent, clarify whether some provided written consent and others verbal, or if both types of consent were required from all subjects. If both were required, it is unnecessary to specify verbal consent. ·
Response: 5 The participants provided written consent. This information was added in the manuscript following the reviewer´s suggestion.
Comment 6:
Detail the equipment and tools used. ·
Response 6: Detail the equipment and tools used have been added.
Comment 7:
Specify whether there were multiple evaluators or if tasks were performed independently by each evaluator. This could be seen as a positive factor if intra-evaluator practice does not interfere; otherwise, it may highlight the need for experience in the area.
Response 7: A professional with 10 years of experience using ultrasound images of the musculoskeletal system used a Logiq S7 with a 8 MHz linear probe (GE Healthcare, Milwaukee,WI, USA) to carry out all measurements.
Comment 8:
Finally, it is unclear whether a total of 23 or 46 subjects were evaluated, as there is a discrepancy between the text and the flowchart.
Response 8: A total of 23 subjects were evaluated. The authors used the non-dominant limb as the control group and the dominant limb as the intervention group. That is why there are 23 participants in each group in bothas the section “2.4 Allocation” and the flow-diagram.
Thank you for your thorough review and insightful comments. We greatly appreciate the time and effort you have taken to provide valuable feedback, which has helped us to improve the clarity, accuracy, and clinical relevance of our manuscript. We have carefully addressed each of your suggestions and believe these revisions have strengthened the overall quality of the paper.
We hope that the revised version meets your expectations and that it is now suitable for publication.
Thank you once again for your constructive input.
Reviewer 2 Report
Comments and Suggestions for Authors
Abstract:
CRMR: The authors need to define this acronym in the abstract, i.e. Capacitive-resistive monopolar radiofrequency (CRMR)
Introduction:
1. “Given the high prevalence of PF, preventive strategies are crucial. Strengthening the intrinsic muscles of the foot, using orthotic insoles, wearing appropriate footwear, and performing stretching exercises are commonly recommended to reduce the incidence of plantar fasciitis [5]. However, despite these preventive measures, many patients continue to experience symptoms, underscoring the need for effective treatment options [6].”
This is irrelevant to the purpose of this study and should be eliminated.
2. Statements of purpose, lines 69-80: “Our hypothesis is that the application of 448 kHz CRMR can induce lasting changes in the elastic properties of the PF, which can be measured through SEL. These changes would enhance our understanding of the physiological responses that occur in healthy PF tissue over time following CRMR intervention, while also supporting the use of SEL as a reliable tool for assessing tissue quality.” And “The aim of this study is to evaluate the changes in the viscoelastic properties of the PF, as measured by SEL, following six sessions of 448 kHz CRMR (twice weekly) over a three-month follow-up period in healthy active subjects. Measurements will be taken at six time points: immediately before the intervention (T0), after two weeks of CRMR intervention program(T1), immediately after the end of the intervention program(T2), two weeks after the intervention (T3), one month after the intervention (T4), and three months after the intervention (T5).”
This is the most important statement in the entire paper. It is not clear. It looks like the authors are studying through strain elastography the quality of the tissue in normal subjects, before and after the application of CRMR. They claim that this will “enhance our understanding of the physiological responses that occur in healthy plantar fascia tissue over time while also supporting the use of strain elastography as a reliable tool for assessing tissue quality”. Then they briefly summarize what they are going to do. I am not really sure how this is useful. The authors should persuade us why this paper should be published. It is a very general thing to say that “another research paper will enhance our understanding of the physiological responses that occur in healthy tissue over time”. We need to know why this is really useful and especially clinically. For example: How will this be used in the clinical practice and if this is a first step what is this helpful for? How this will change the way we practice? I am really confused about the usefulness of this paper and what the authors are trying to show that we can use in our everyday practice. Unless they clarify their statement of purpose I'm not sure that this effort is worthy of publication. Furthermore, the authors state that “CRMR can induce lasting changes in tissue elasticity”, whereas the change the changes only lasted 1 month in their study, therefore this is an overstatement.
Some general comments, which may help the authors are presented herein.
Materials and methods:
- Line 106: Allocation: why are you using the non dominant as the control group and the dominant as the intervention group? Wouldn't it be preferable to have two groups of patients and consider the dominant as the intervention and the non dominant as the control for Group 1 and have another group (Group 2) that has the inverse? That would make your results more objective.
- Line 121: The authors shoot the describe which company manufacturs this “INDIBA Activ 8” equipment.
- The control group is not described here. Did they receive any intervention? Or did the authors just did the SEL? It should have a separate subheading 2.6.2, then 2.6.3 for the SEL measurements.
Results:
- Line 158-163: this describes the participants and the intervention and therefore it belongs to the materials and methods sections and should not be misplaced in the results section.
Discussion:
- It would be an overstatement to say that CRMR can induce lasting changes in tissue elasticity, as the changes only last 1 month.
- Lines 218-219: Sure, but how is this clinically relevant?
Author Response
Dear Reviewer,
We greatly appreciate your detailed review and have made significant modifications to the manuscript to clarify and emphasize the clinical relevance of our findings, fully addressing your observations. Below, we outline the main changes made and how they enhance the study’s focus and contribution.
Comment Abstract:
CRMR: The authors need to define this acronym in the abstract, i.e. Capacitive-resistive monopolar radiofrequency (CRMR)
Response: Thank you. This has been amended
Introduction:
Comment 1. “Given the high prevalence of PF, preventive strategies are crucial. Strengthening the intrinsic muscles of the foot, using orthotic insoles, wearing appropriate footwear, and performing stretching exercises are commonly recommended to reduce the incidence of plantar fasciitis [5]. However, despite these preventive measures, many patients continue to experience symptoms, underscoring the need for effective treatment options [6].”
This is irrelevant to the purpose of this study and should be eliminated.
Response 1: This part of the text has been deleted as suggested.
Comment 2. Statements of purpose, lines 69-80: “Our hypothesis is that the application of 448 kHz CRMR can induce lasting changes in the elastic properties of the PF, which can be measured through SEL. These changes would enhance our understanding of the physiological responses that occur in healthy PF tissue over time following CRMR intervention, while also supporting the use of SEL as a reliable tool for assessing tissue quality.” And “The aim of this study is to evaluate the changes in the viscoelastic properties of the PF, as measured by SEL, following six sessions of 448 kHz CRMR (twice weekly) over a three-month follow-up period in healthy active subjects. Measurements will be taken at six time points: immediately before the intervention (T0), after two weeks of CRMR intervention program(T1), immediately after the end of the intervention program(T2), two weeks after the intervention (T3), one month after the intervention (T4), and three months after the intervention (T5).”
This is the most important statement in the entire paper. It is not clear. It looks like the authors are studying through strain elastography the quality of the tissue in normal subjects, before and after the application of CRMR. They claim that this will “enhance our understanding of the physiological responses that occur in healthy plantar fascia tissue over time while also supporting the use of strain elastography as a reliable tool for assessing tissue quality”. Then they briefly summarize what they are going to do. I am not really sure how this is useful. The authors should persuade us why this paper should be published. It is a very general thing to say that “another research paper will enhance our understanding of the physiological responses that occur in healthy tissue over time”. We need to know why this is really useful and especially clinically. For example: How will this be used in the clinical practice and if this is a first step what is this helpful for? How this will change the way we practice? I am really confused about the usefulness of this paper and what the authors are trying to show that we can use in our everyday practice. Unless they clarify their statement of purpose I'm not sure that this effort is worthy of publication. Furthermore, the authors state that “CRMR can induce lasting changes in tissue elasticity”, whereas the change the changes only lasted 1 month in their study, therefore this is an overstatement.
Response 2:
Dear reviewer, thank you for your comment, which has been taken into account in both the introduction and discussion. We have expanded the Clinical Significance of the Results to provide a detailed explanation that this is the first study investigating the effects of CRMR on plantar fascia (PF) elasticity in healthy subjects. This represents a crucial foundational step for understanding physiological responses in a healthy population before applying CRMR in pathological conditions such as plantar fasciitis. By adding this clarification, we underscore that the study’s purpose goes beyond exploring a new CRMR application; it is about establishing baseline knowledge on PF elasticity changes as groundwork for future clinical applications.
Clinical Relevance and Practical Implications: In response to your comments, we have emphasized the clinical utility of elasticity changes in high-demand PF areas, such as the medial calcaneal process. We explain that these results suggest that CRMR could improve the load-bearing capacity of the PF, which is particularly relevant for active populations prone to overuse, like athletes. Additionally, we now highlight that the one-month duration of effects is clinically meaningful, potentially reducing session frequency compared to other therapies, thus decreasing treatment burden for patients and supporting sustained improvements in tissue function. This expands our clinical conclusions, providing a clearer perspective on how these findings could be applied in practice.
Use of SEL as an Assessment Tool: We have included a detailed rationale for using strain elastography (SEL) as a non-invasive and quantitative method for monitoring tissue quality over time. The discussion now explores how SEL not only allows precise and repeatable assessments of PF elasticity but also establishes a reference range that could be useful for tracking pathological conditions in future studies. This directly addresses the reviewer’s question regarding the utility of SEL in the study and its clinical applicability.
Preventive Perspective and Future Research Potential: We expanded the discussion on the preventive role of CRMR in healthy individuals and at-risk populations, such as runners and those standing for prolonged periods. By enhancing PF elasticity in these individuals, CRMR could reduce the risk of plantar fasciitis and other degenerative conditions. We further discuss how these findings provide a strong basis for future research that evaluates CRMR effects in subjects with pathological conditions, exploring long-term interventions and specific treatment protocols.
Therefore, in the revised version, we have ensured that the study’s relevance and purpose are clearly articulated from the introduction and discussion, highlighting its pioneering and clinically significant nature. These additions and clarifications improve the manuscript’s cohesiveness and emphasize how our findings offer not only a scientific foundation for further research but also a practical and preventive clinical proposal.
Comment 3:
Some general comments, which may help the authors are presented herein.
Materials and methods:
- Line 106: Allocation: why are you using the non dominant as the control group and the dominant as the intervention group? Wouldn't it be preferable to have two groups of patients and consider the dominant as the intervention and the non dominant as the control for Group 1 and have another group (Group 2) that has the inverse? That would make your results more objective.
Response 3: Thank you for your comment regarding the allocation design. The authors chose to use the dominant limb as the intervention group and the non-dominant limb as the control within the same participants to minimize the impact of external factors that could confound the results. This approach allows us to control for individual variables that might influence treatment response, such as nutrition, mental health, and lifestyle habits. By using the same participants for both the intervention and control, we ensure that any changes in the elastic properties of the plantar fascia can be more accurately attributed to the capacitive-resistive monopolar radiofrequency (CRMR) treatment, reducing the influence of external variability.
Additionally, this design provides a methodological advantage: if changes in the elasticity of the plantar fascia were also observed in the untreated limb (i.e., the contralateral side), this might suggest a systemic or reflex response to CRMR intervention. However, in this study, we did not observe significant changes in the contralateral limb, supporting that the effects of CRMR are localized to the treated limb.
We appreciate your suggestion of using independent intervention and control groups, which could enhance the objectivity of the study. We acknowledge that the chosen design is a potential limitation, and therefore we have included this consideration in the study’s limitations.
Comment 4:
- Line 121: The authors shoot the describe which company manufacturs this “INDIBA Activ 8” equipment.
Response: Thank you for your observation regarding the manufacturer of the "INDIBA Activ 8" equipment mentioned in line 121. We acknowledge the importance of providing complete information about the devices used in our study. The "INDIBA Activ 8" is produced by INDIBA Group, a company specializing in radiofrequency technology for medical and aesthetic applications. We will include this detail in the revised manuscript to ensure clarity and transparency.
- The control group is not described here. Did they receive any intervention? Or did the authors just did the SEL? It should have a separate subheading 2.6.2, then 2.6.3 for the SEL measurements.
Response 4: The authors have added a separate subheading 2.6.3 where the control group non-intervention was described.
Comment 5:
Results:
- Line 158-163: this describes the participants and the intervention and therefore it belongs to the materials and methods sections and should not be misplaced in the results section.
Response 5: We have displaced this paragraph at the materials and methods section following the reviewer's indication.
Comment 6:
Discussion:
- It would be an overstatement to say that CRMR can induce lasting changes in tissue elasticity, as the changes only last 1 month.
Response 6: Thank you for your observation regarding the duration of changes induced by CRMR. We agree that stating "lasting changes" may imply a longer duration than observed in this study. To clarify, we have revised the language in the manuscript to reflect that the CRMR-induced changes in tissue elasticity persisted specifically for up to one month in our study. This duration is nonetheless significant in a clinical context, as it suggests that even short-term CRMR interventions can produce measurable improvements in tissue elasticity for a period that may reduce the need for frequent treatment sessions.
In light of your comment, we have adjusted the wording in the text to more accurately represent the findings, acknowledging that further research is needed to explore how the duration of these effects might be extended or maintained with adjusted CRMR protocols.
Comment 7:
- Lines 218-219: Sure, but how is this clinically relevant?
Response 7: Thank you for your comment regarding the clinical relevance of our findings. For us, the clinical relevance lies in the potential to develop targeted treatment protocols that can be tailored to specific tissue types and mechanical demands. Based on the observed elasticity changes in high-load regions of the plantar fascia following CRMR application, this knowledge could support personalized interventions aimed at both injury prevention and rehabilitation. These protocols could be adjusted in terms of treatment frequency, intensity, and duration according to the specific needs of each patient, whether to optimize tissue resilience in healthy individuals or to enhance functionality and reduce pain in patients with plantar fascia-related pathologies. By establishing these tailored approaches, we hope to apply CRMR more effectively in clinical settings, enhancing tissue resilience in at-risk populations and offering a non-invasive treatment option for managing degenerative conditions of the plantar fascia.
We have clarified this point in the revised manuscript and included it in the Discussion section. Also in limitations sections we note the importance of further research to optimize CRMR parameters and protocol development for different patient needs. This addition emphasizes how our findings could serve as a foundation for future clinical applications and protocol standardization.
We trust these modifications improve the manuscript’s clarity and value, addressing all aspects raised in your feedback. We remain open to any further suggestions.
Round 2
Reviewer 1 Report
Comments and Suggestions for Authors
The authors have incorporated most of the observations made in the review of the previous version, resulting in a notable improvement in the clarity, rigor, and clinical relevance of the work. However, I believe certain aspects still need to be clarified and clearly reflected in the manuscript.
Specifically, they should indicate that the diagnosis of the pathology was conducted via ultrasound, measuring thickness and establishing the diagnosis based on this quantitative value, while also clarifying the protocol used. Including a bibliographic justification of this procedure would be beneficial.
Additionally, they mention that the diagnosis was made by an expert with over 10 years of experience, but they also state that the interventions were carried out by multiple specialists. Were all patients diagnosed by the same professional before being referred to other specialists? These issues should be clarified and discussed, addressing both positive and negative aspects in the discussion section, as suggested in the previous review
Reviewer 2 Report
Comments and Suggestions for Authors
All the comments were adequately addressed.
